# Broadband thermal imaging using meta-optics

Luocheng Huang[1], Zheyi Han[1], Anna Wirth-Singh[2], Vishwanath Saragadam [3], Saswata Mukherjee[1], Johannes E. Fröch[1,2], Quentin A. A. Tanguy[1], Joshua Rollag[4,5], Ricky Gibson [5], Joshua R. Hendrickson [5], Philip W. C. Hon[6], Orrin Kigner [6], Zachary Coppens[7], Karl F. Böhringer [1,8], Ashok Veeraraghavan [3] & Arka Majumdar [1,2] ✉

Subwavelength diffractive optics known as meta-optics have demonstrated the potential to significantly miniaturize imaging systems. However, despite impressive demonstrations, most meta-optical imaging systems suffer from strong chromatic aberrations, limiting their utilities. Here, we employ inverse-design to create broadband meta-optics operating in the long-wave infrared (LWIR) regime (8-12 $\mu m$). Via a deep-learning assisted multi-scale differentiable framework that links meta-atoms to the phase, we maximize the wavelength-averaged volume under the modulation transfer function (MTF) surface of the meta-optics. Our design framework merges local phase-engineering via meta-atoms and global engineering of the scatterer within a single pipeline. We corroborate our design by fabricating and experimentally characterizing all-silicon LWIR meta-optics. Our engineered meta-optic is complemented by a simple computational backend that dramatically improves the quality of the captured image. We experimentally demonstrate a six-fold improvement of the wavelength-averaged Strehl ratio over the traditional hyperboloid meta-lens for broadband imaging.

Long-wavelength infrared (LWIR) imaging is a critical key technology for non-contact thermography, long-range imaging, and remote sensing, with applications spanning consumer electronics to defense[1]. High resolution imaging in LWIR typically requires bulky and precisely engineered refractive surfaces, which ultimately add to the overall volume and weight of the imaging system, especially for high numerical aperture (NA) optics. To reduce the size and weight of LWIR imagers, diffractive optical elements such as multilevel diffractive lenses (MDL)[2] have been used as an alternative to traditional refractive optics. However, the use of MDLs has been limited by the complexity of the multi-layer fabrication and the large periodicity, resulting in higher order diffraction. Sub-wavelength diffractive optics, also known as meta-optics, have recently generated strong interest to spatially modulate the phase, amplitude, and polarization of the incident wavefront. These consist of scatterers, which are placed on a sub-wavelength periodic lattice to avoid any higher order diffraction, whereas the active layer thickness of the device corresponds to the height of the scatterers[3–16]. Due to the drastic thickness and weight reduction in meta-optics compared to refractive lenses, LWIR meta-optical imaging under ambient light conditions have been recently reported[17–19]. However, until this point, the captured image quality remained relatively poor compared to refractive lenses, primarily due to strong chromatic aberrations.

Some drawbacks of meta-optics are inherited from their diffractive counterparts, the most significant one being their strong axial chromatic aberration. This happens because the phase wrapping

[1]Department of Electrical and Computer Engineering, University of Washington, Seattle, WA, USA. [2]Department of Physics, University of Washington, Seattle, WA, USA. [3]Department of Electrical Engineering, Rice University, Houston, TX, USA. [4]KBR, Inc., Beavercreek, OH, USA. [5]Sensors Directorate, Air Force Research Laboratory, Wright-Patterson AFB, OH, USA. [6]NG Next, Northrop Grumman Corporation, Redondo Beach, CA, USA. [7]CFD Research Corporation, Huntsville, AL, USA. [8]Institute for Nano-Engineered Systems, University of Washington, Seattle, WA, USA. ✉e-mail: arka@uw.edu

condition for different wavelengths is met at different radii[13,20]. The dispersion engineering approach can ameliorate this axial chromatic aberration to some degree, but ultimately faces fundamental limitations explicitly set by the achievable group delay and group delay dispersion[21] for large aperture meta-optics. To overcome this fundamental limitation and enhance the image quality, chromatic aberrations in a large-aperture meta-optics can be mitigated via computational imaging using forward designed meta-optics, that exhibit extended depth of focus (EDOF) properties, as demonstrated in the visible regime[22,23].

However, for forward designed meta-optics an optical designer has to strongly rely on experience and intuition about the functionality of the meta-optics. As such, this method does not provide a clear path to further improve the imaging quality. Unlike forward design, inverse design approaches define the performance of the optical element via a figure of merit (FoM), and computationally optimize their structure or arrangement to maximize the respective FoM. The inverse design methodology has been very successful in creating non-intuitive yet functional meta-optics[24–26], including EDOF lenses for broadband imaging in the visible. A further refinement and extension of this approach is end-to-end design, where the meta-optics and computational backend are co-optimized with a FoM defined by the final image quality[27]. While such an approach takes the entire system into account, the downside is that we often lack the insight into how and why the optic performs well. This can be detrimental when translating designs from the visible domain (with ample training data) to the thermal domain (with paucity of training data). As such, a new design paradigm is required for meta-optical imagers, which provides intuition on why such meta-optics can perform broadband imaging.

In this paper, we report a 1 cm (~1000λ) aperture, f/1 (numerical aperture (NA): 0.45) broadband polarization-insensitive LWIR meta-optics, designed by a multi-scale optimization technique that

maximizes the wavelength-averaged volume under the modulation transfer function (MTF) of the meta-optic. This method is multi-scale as distinct computational methods are used to calculate the electric fields at different length scales. The meta-optics are implemented in an all-silicon platform and we demonstrate that the captured images exhibit superior performance over traditional hyperboloid metalens. Unlike previous works, which relied either on local engineering of meta-atoms (dispersion engineering) or global engineering of the phase-mask, our optimization technique employs both meta-atom and phase-mask engineering. We show that such an extended approach clearly outperforms a phase-mask engineering only approach.

## Results

### Inverse design framework

The key to achieve meta-optical broadband LWIR imaging is a fully-differentiable inverse design framework that optimizes the binary structures (i.e., binary permittivity distribution) of the meta-atoms to achieve focusing of the desired wavelengths (Fig. 1a). However, designing a large-area (~$10^3\lambda$) meta-optic is a computationally prohibitive problem, and full-wave simulation is not possible. Therefore, phase-mask optimization is generally used, but the scatterer-to-phase mapping on a sub-wavelength scale is not differentiable and a look-up-table based approach does not work for such an optimization methodology. In the past, polynomial proxy functions have been employed to connect scatterers to the corresponding phase[27], but were limited to only a monotonic relation. However, for broadband operations meta-atoms with a large phase diversity with multiple phase wrappings are required but suffer from multiple resonances at various wavelengths.

We solve this problem by utilizing a deep neural network (DNN)-based surrogate model, that maps the geometrical parameters ($\vec{p}$) of the meta-atom to its complex transmission coefficient $\hat{E}(\vec{p}, \lambda)$. The utilization of DNNs enable modeling highly complex functions, which

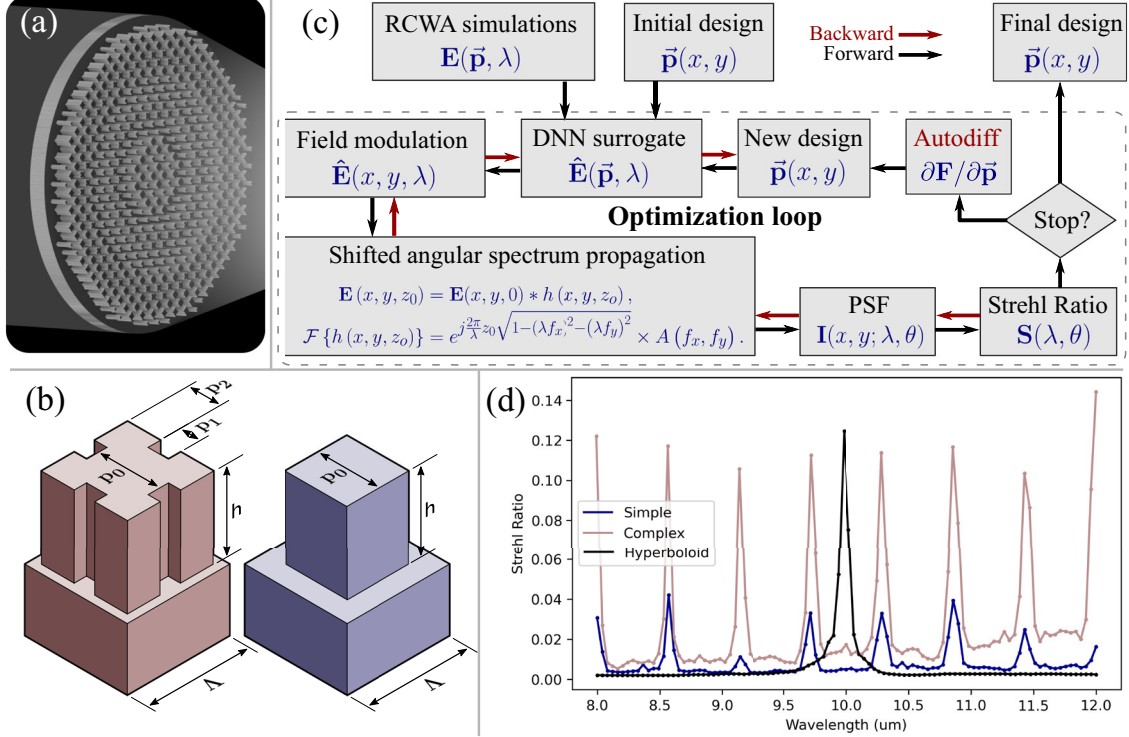

**Fig. 1 | Design methodology to create the broadband LWIR meta-optics. a** Our objective is a LWIR meta-optic that focuses broadband light at the same focal plane. **b** Parameterization of the complex (red) and the simple (blue) scatterers. For the complex scatterers, three variable parameters control the binary profile, namely, **p₀, p₁,** and **p₂.** This method of parameterization ensures 4-fold symmetry of the

geometry, thus ensuring polarization-insensitivity. The height **h** is kept constant at 10 μm. The meta-atoms sit on a square lattice with a periodicity **Λ** set to 4 μm. **c** Block diagram of the optimization routine. **d** The simulated Strehl ratio of the simple, complex, and hyperboloid meta-optic where perturbations are introduced to simulate the imperfections inherent in the fabrication process.

are well suited for broadband meta-optics. To emphasize this point, we compare meta-atoms with different degrees of parameterization: the simple scatterers are defined by one design parameter (the width of the pillar), whereas the complex scatterers have three parameters, see Fig. 1b. For each type of meta-atom, we sampled the parameters $\vec{p}$ uniformly for each feature dimension and simulated it using rigorous coupled wave analysis (RCWA)[28], across different wavelengths $\lambda$. The simulated meta-atom library is then taken as the training data set for the DNN that maps the parameterized features to the phase modulation. The RCWA simulations and the DNN fitting are pre-computed before the optimization loop, shown in Fig. 1c. Although we are currently limited to parameterized meta-atoms, with the rapid progress of machine learning-enabled meta-optics design, we anticipate voxel-level meta-atom engineering will be possible in the near future. Also, our design currently relies on the local phase approximation, which is known to provide lower efficiency for meta-optics. However, recently a physics-inspired neural network has been used to design meta-optics without making such approximation and can be readily adapted for the scatterer-to-field mapping[29].

We define the FoM by computing the normalized MTFs first as $\hat{M}(k_x, k_y, \lambda, \theta)$, where $k_x, k_y$ are defined as the spatial frequencies in the x and y direction, $\lambda$ is the optical wavelength, and $\theta$ signifies the incident angle (more details in Methods). The modified Strehl Ratio $\mathbf{S}$ can then be calculated using the following expression:

$$
\begin{aligned}
\mathbf{S}(\lambda,\theta) &= \int_{-k_{y1}}^{k_{y1}} \int_{-k_{x1}}^{k_{x1}} \hat{M}\left(k_x, k_y; \lambda, \theta\right) dk_x dk_y \\
&= \sum_i \sum_j \hat{M}\left(k_x(i), k_y(j); \lambda, \theta\right) \Delta k_x \Delta k_y
\end{aligned}
\tag{1}
$$

Here, $k_{x1}$ and $k_{y1}$ denote the cutoff spatial frequencies in the x and y direction. Using the modified Strehl Ratio, the FoM $\mathbf{F}$ is computed as:

$$
\mathbf{F} = \log\left( \prod_i \prod_j S(\lambda_i, \theta_j) \right)
\tag{2}
$$

This FoM can be interpreted as the wavelength and incidence angle-averaged volume under the MTF surface, and we therefore term this optimization routine as MTF engineering. We emphasize that our FoM is maximized when all individual $S(\lambda_i, \theta_j)$ are identical, thus constraining our meta-optics to have a uniform performance for the specified wavelengths without explicitly defining uniformity as an optimization criteria. Since the entirety of the forward computation of the FoM is implemented using differentiable operations, the gradient of the FoM with respect to the geometry parameters, i.e. $\partial \mathbf{S}/\partial \vec{p}$, can be readily obtained using automatic differentiation through chain rule in the direction indicated by the red arrows in Fig. 1c. A stochastic gradient descent algorithm then optimizes the structural parameters $\vec{p}$. The optimization routine halts when the value of FoM converges, yielding the final meta-optic design.

## Meta-optics design

Using the MTF engineering framework, we designed the broadband LWIR meta-optics. We opted for an all-silicon platform, which is composed of silicon pillars on a silicon substrate, to simplify the fabrication process. We note that while silicon does absorb some of the LWIR light, we still expect about 80% light transmission.

In this study, we designed two different broadband meta-optics, each with a unique scatterer archetype shown in Fig. 1b. Both archetypes were parameterized to ensure fourfold symmetry, which leads to polarization insensitivity. To ensure high transmission efficiency, we retained only those meta-atoms which have transmission exceeding 60%. Additionally, we designed a hyperboloid metalens, based on a forward design approach[23], possessing similar height and periodicity, to serve as a baseline for comparison. All designed meta-optics have a nominal focal length of 1 cm and a numerical aperture of 0.45. In our simulations, the optimized broadband meta-optics displayed significantly larger wavelength-averaged Strehl Ratios−0.045 for the meta-optics with complex scatterers and 0.018 for those with simple scatterers, as compared to 0.0075 for the forward-designed hyperboloid metalens. We can qualitatively explain the higher Strehl ratio with complex scatterers, as they can provide higher phase diversity, which will help to satisfy the phase distribution for different wavelengths. Essentially, such complex scatterers help to achieve a similar effect of dispersion engineering to achieve broadband performance. The simulations were, however, limited to 8 optimized wavelengths spanning from 8 to 12 μm due to memory constraints. Fig. 1d depicts the simulated Strehl ratios of the optics described above in relation to the input wavelength. For these simulations, individual meta-atoms were simulated using RCWA, while DNN mapping was utilized solely for optimization. To mimic fabrication imperfections, we introduced normally distributed perturbation into each meta-atom's design parameters. Remarkably, the complex meta-optic design yielded Strehl ratios at eight sampled wavelengths that are comparable to the Strehl ratio at a single operational wavelength of the hyperboloid metalens. We note that, in these simulations, we added normally distributed perturbation into each meta-atom's design parameters, simulating fabrication imperfections, resulting in a less-than-perfect Strehl ratio for the hyperboloid metasurface at the desired wavelength. More details on the effect of fabrication imperfections on the properties of meta-atoms are reported in the Supplementary Materials. We emphasize, however, while fabrication imperfections will affect the meta-optics captured images, the use of a computational backend can provide additional robustness in the overall imaging performance.

Although the spectral regions between the sampled wavelengths exhibit relatively lower Strehl ratios compared to the peak values, these ratios for the non-sampled wavelengths still remain significantly larger than those of the hyperboloid lens at the same wavelengths. As such, when averaged over all the wavelengths of interest, we still obtain a six-fold improvement for the average Strehl ratio. Our experimental results demonstrate that, despite such polychromatic behavior, it is possible to capture images under broadband ambient thermal radiation. This highlights the practicality and adaptability of our broadband meta-optic designs in real-world scenarios.

## Fabrication

We fabricated the LWIR meta-optics in an all-silicon platform, with a set of devices shown in Fig. 2a. We pattern the Si wafer with laser direct-write lithography and etch the patterns using deep reactive ion etching (details in Methods). We note that patterning can also be accomplished with a mask aligner – therefore our all-silicon platform can be adapted to large scale foundry processes. Scanning electron microscope images of the fabricated complex and simple meta-optics are depicted in Fig. 2b and c, respectively.

## Imaging

We first characterized the PSF of the fabricated lenses (results are provided in the Supplementary Materials), and observed less variation of the PSFs across the LWIR range as expected. To conclusively demonstrate that the MTF-engineered optics deliver superior imaging quality compared to the forward-designed metalens, we evaluated their respective imaging performance under controlled conditions (see Methods for details). As illustrated in Fig. 3, the images captured using the MTF-engineered meta-optic with complex scatterers exhibit the highest level of clarity and detail. Stripe elements of an imaging target can be clearly distinguished and well separated, emphasizing the overall crispness of the images.

To highlight the broadband capability of the MTF-engineered meta-optics, we captured images using both 10 μm and 12 μm band-pass filters (each with a linewidth of 500 nm). We ensured a consistent

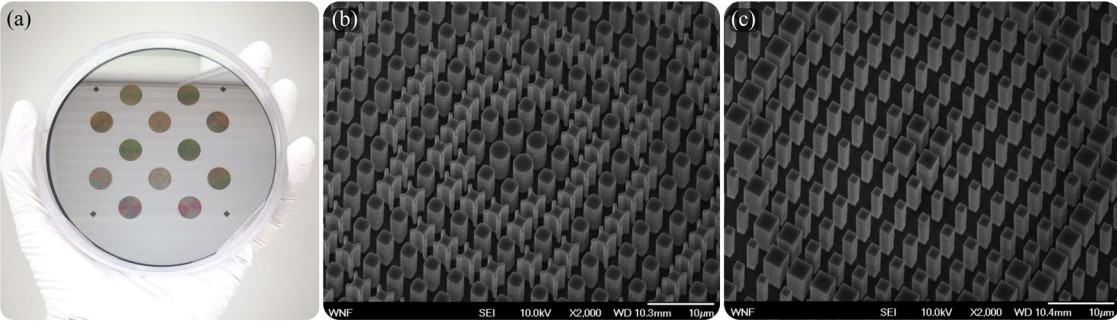

**Fig. 2 | Images of the fabricated meta-optics. a** Camera image of a fabricated wafer with several meta-optics. SEM images of the meta-optics with **b** complex scatterers and **c** simple scatterers.

distance between the meta-optics and the imaging sensor, maintaining a nominal operating wavelength of $11\,\mu m$. The resultant imaging outputs, which are presented without any denoising, are displayed at the bottom of each sub-figure in Fig. 3. Notably, the MTF-engineered meta-optic with complex scatterers outperforms both the simple scatterers and the forward-designed optics for both the line and husky targets. Additionally, the MTF-engineered meta-optic with simple scatterers demonstrates superior performance over the forward-designed hyperboloid metalens, as evidenced by a clearer image and higher peak signal-to-noise ratio (PSNR) for both patterns in both the $10\,\mu m$ and $12\,\mu m$ cases. A comprehensive list of PSNR values can be found in Supplementary Materials.

To further emphasize the capabilities of the MTF-engineered LWIR meta-optics, we performed imaging in-the-wild under both indoor and outdoor ambient daylight conditions, shown in Fig. 4. In each case, we captured a single image using a FLIR A65 sensor, and performed a numerical deconvolution (details in the Methods and Supplementary Materials). The MTF-engineered meta-optic with complex scatterers and forward-designed meta-optics were tested against a refractive lens for the ambient imaging, seen on Fig. 4a. First, we imaged a truck parked outdoors shown in Fig. 4b. The vehicle details are clearly visible such as the front grill, as well as background details, such as the windows on the top right corner of the image, and the stairs on the left side of the image captured using MTF-engineered meta-optic. In contrast, the hyperboloid image suffers from strong glowing artifacts in the center and fails to recover sharp details. Fig. 4c shows images of a parked car. As with the previous example, the details are clearly visible, including the tires on the car, and the branches of the tree in the background with the MTF-engineered lens. Fig. 4d shows image of a person with hands stretched out. The difference between MTF-engineered lens and hyperboloid metalens is distinctly visible here. The sharp image features are clearly visible in the MTF-engineered lens, including variable heat patterns on the shirt. In contrast, the image formed by the hyperboloid lens is noisy with no clear features on the person.

## Discussion

In this work, we have devised an inverse design methodology for broadband imaging meta-optics, guided by readily translatable, intuitive, and universal objective function, given by the volume of a multichromatic MTF surface. Utilizing this MTF-engineering approach, we achieve a sophisticated, yet easily fabricable, large-aperture, broadband LWIR meta-optic, suitable for in-the-wild imaging under ambient temperature conditions. We have experimentally verified this framework and demonstrated polarization-insensitive broadband LWIR meta-optics with a diameter of 1 cm and NA of 0.45. While previous works based on forward designed LWIR meta-optics have demonstrated imaging capabilities[23], they fell short in resolving fine features

due to strong chromatic aberrations. In contrast, our MTF-engineered meta-optics show significantly improved performance over a broadband spectral range and narrowband imaging capabilities for wavelengths outside the center wavelength.

We further elucidate, both in simulation and experiment, how a significant performance enhancement can be achieved for MTF-engineered meta-optics if we consider more structural degrees of freedom. Such complex parameterization of the meta-optics broadens the solution space during the optimization process, thereby increasing the likelihood of achieving an improved FoM. This provides a clear pathway for future designs to leverage a performance boost by employing a higher degree of parameterization for the meta-optic scatterer, combined with large-scale optimization of the meta-optic. We note that previous works primarily employed either meta-atom engineering or phase-mask optimization, often overlooking potential synergistic effects. We demonstrate unequivocally that by utilizing structural diversity along with global phase-mask engineering, a six-fold performance improvement can be achieved. However, a clear downside is that the sampling complexity increases exponentially with the number of structural parameters. Additionally, fabrication resolution requirements become more stringent with the increased complexity of the meta-atoms. Despite these challenges, they can potentially be overcome by using a more clever parameterization of the meta-atom, similar to what has been achieved in dispersion engineering approaches. We note that, unlike many other works, we have not directly emphasized the need for high efficiency. In meta-optics community, historically two different efficiencies have been reported: transmission and focusing efficiency. The transmission efficiency indicates how much light gets transmitted through the optic, and focusing efficiency determines how much of the transmitted light gets into the focused region. The focusing efficiency is somewhat arbitrarily defined, and has almost no counterpart for refractive optics. Hence, in our work, we do not optimize focusing efficiency. However, our modified Strehl ratio implicitly takes account of the focusing efficiency. If the light is not tightly confined, and a large amount of scattered light is present, we will have a large DC component in the MTF which will reduce the average Strehl ratio. Thus, our MTF-engineering method indirectly optimizes the focusing efficiency. To ensure high transmission efficiency, we pre-select the meta-atoms with high transmission coefficient.

Overall, the developed MTF-engineering framework provides a universal methodology for creating large-area broadband meta-optics. Our imaging and characterization results clearly demonstrate the advantages of employing the MTF-engineering methodology over the traditional forward design. We showcase that the MTF-engineered flat optics open a new avenue for miniaturizing LWIR imaging systems, with potential applications in unmanned aerial vehicles, thermal scopes, and perimeter security.

| Hyperboloid Metalens | MTF Engineered (simple scatterers) | MTF Engineered (complex scatterers) |
|---|---|---|

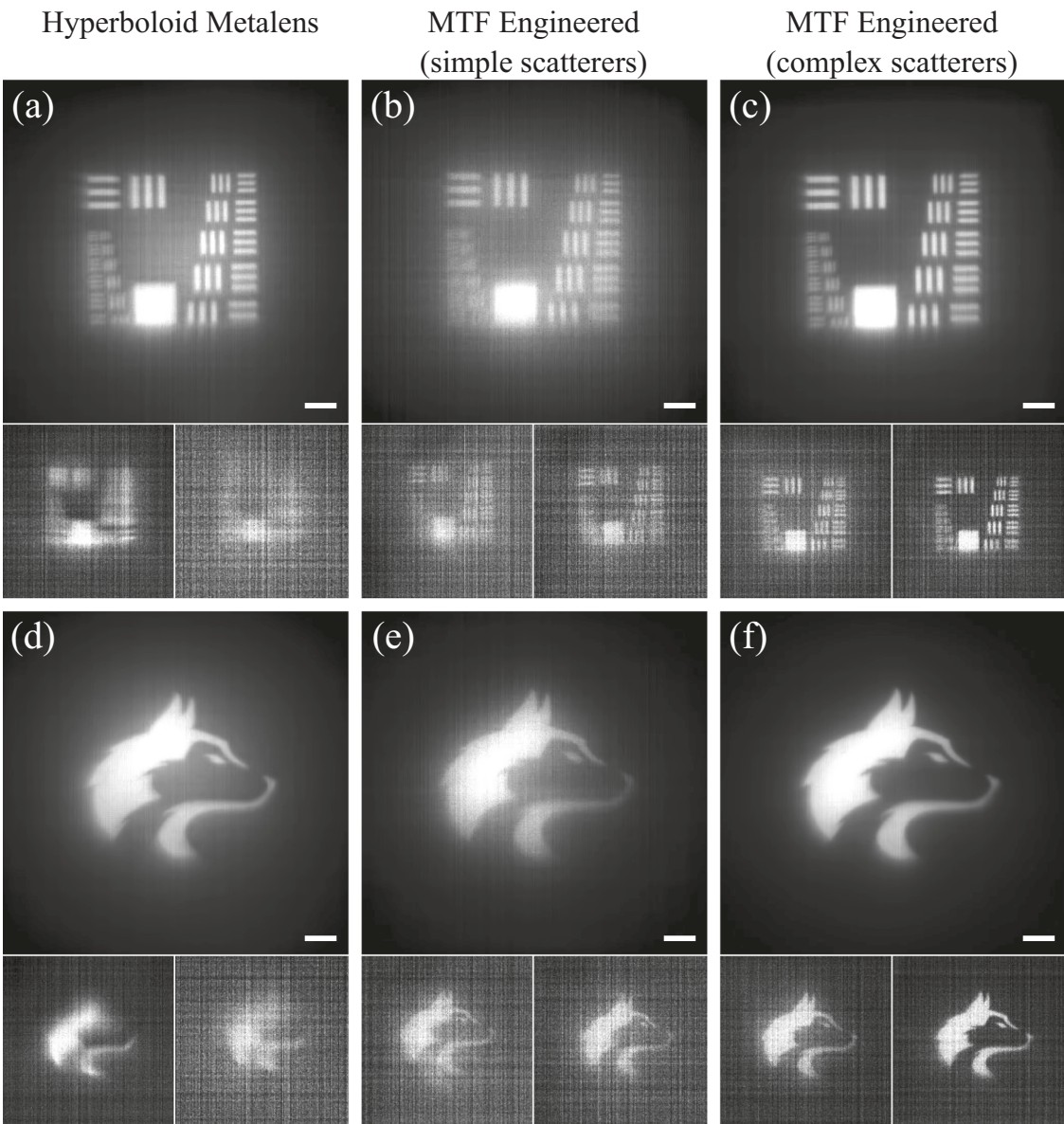

**Fig. 3 | Broadband imaging results in the lab (after computation). a, d** Imaging results for the hyperboloid metalens. **b, e** Results for the MTF-engineered meta-optic with simple scatterers. **c, f** Results for the MTF-engineered meta-optic with complex scatterers. For each subfigure the imaging results are shown without a filter (top), with a $10 \pm 0.25\,\mu m$ band-pass filter (bottom left), and with a $12 \pm 0.25\,\mu m$ band-pass (bottom right). The scale bar is 0.56 mm. Imaging results using two different targets, namely are a modified USAF 1951 pattern (top) and a picture of a husky (bottom) are shown.

## Methods

### MTF calculation

Each meta-atom is assigned a position (x, y) on a Manhattan grid with a periodicity of **Λ** and a height of **h**, whose geometric parameters are $\overrightarrow{\mathbf{p}}$. These geometric parameters are given random assigned initial values within the fabrication limited bounds. The spatial modulation of the meta-optic $\hat{\mathbf{E}}(x,y,\lambda)$ is calculated given the meta-atom position and the wavelengths of the incident fields. The spatial modulation is then multiplied by the incident field (planewave with incidence angle $\theta$) and propagated to the sensor plane centered to the chief ray, using the shifted angular spectrum method[30]. The variable $I(x,y;\lambda,\theta)$ represents the sampled intensities on the sensor plane given all permutations of the wavelengths and angles of incidence of the incident fields. This intensity is the point spread function (PSF) of the imaging system. The MTF ($M$) is computed from the PSF given by $\mathbf{M}(k_x,k_y;\lambda,\theta) = |\mathcal{F}(I(x,y;\lambda,\theta))|$, where $\mathcal{F}$ is the Fourier transform. The diffraction limited MTFs are denoted as $D(k_x,k_y,\lambda)$. The normalized MTF is defined as $\hat{M} = M(k_x,k_y,\lambda)/D(k_x,k_y,\lambda)$.

### Optimization

For the design with the complex scatterer, we used 40 samples per feature dimension in $\overrightarrow{\mathbf{p}}$ while keeping $h$ constant at $10\,\mu m$. Combining with 101 samples of wavelengths $\lambda$, there are totally $40^3 \times 101$ RCWA simulations conducted for training of the DNN model. For the simple scatterer case, we sampled 1000 values for the only structure variable $p_1$ and 101 wavelengths, also keeping $h$ constant at $10\,\mu m$, resulting in $1000 \times 101$ training samples for the surrogate. For training the DNN surrogate model, the Adam optimizer is used with a learning rate of $10^{-3}$ for 1000 epochs. To span a diameter of 10 cm, a total of $2500 \times 2500$ meta-atoms are used with a periodicity of $10\,\mu m$. During optimization, a total of 13 samples of wavelengths are used uniformly spanning from $8\,\mu m$ to $12\,\mu m$. We selected 0° and 10° to be the angles of incidence in our optimization.

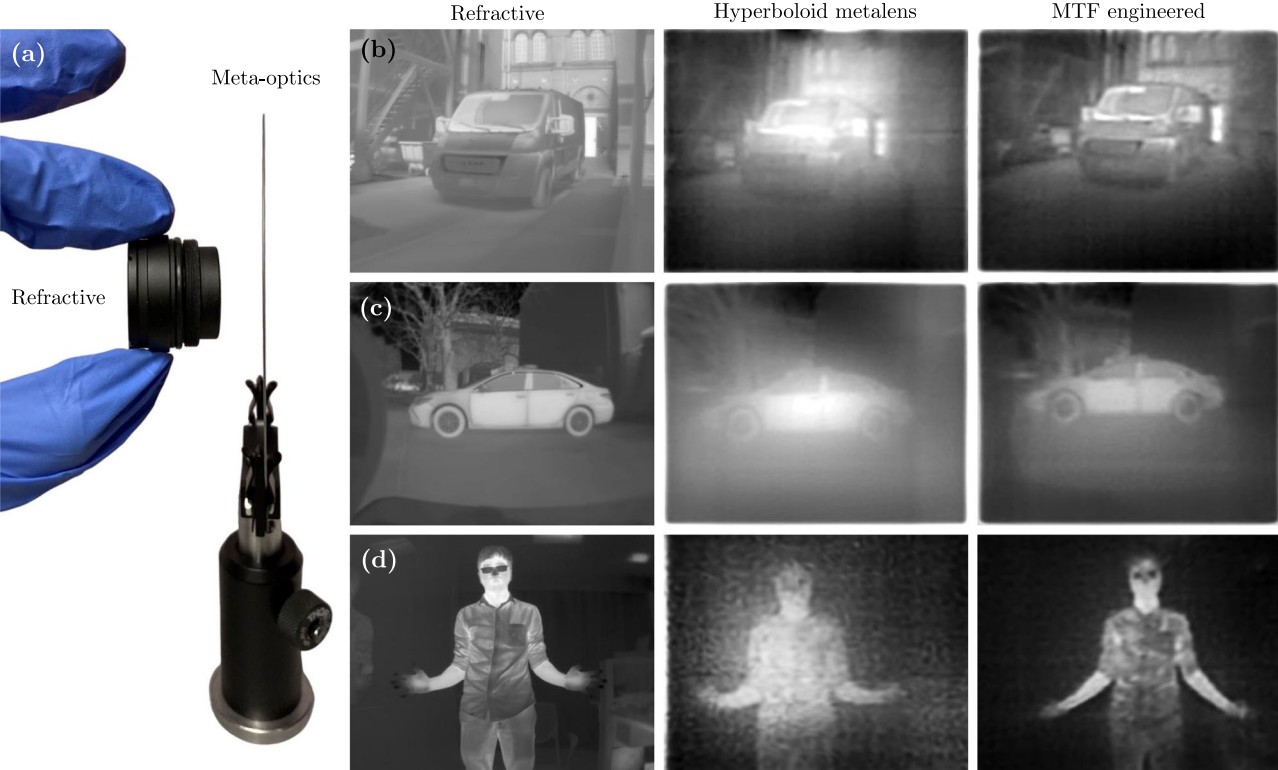

|           | Refractive | Hyperboloid metalens | MTF engineered |

**Fig. 4 | LWIR imaging in-the-wild. a** Thickness comparison between the refractive lens (left) and the meta-optics (right). Three scenes were imaged using the refractive LWIR lens (left), the hyperboloid metalens (middle), and the MTF engineered meta-optic with complex scatterers (right). Scenes **b**, **c** were captured outdoors on a sunny day while **d** was captured indoors.

The DNN surrogate model uses four layers of fully connected layers that have 256 units and hyperboloid tangent as the activation function for each layer. To ensure a differentiable mapping, the fitted surrogate model uses all differentiable operations.

The structure parameters $\vec{p}$ are assigned initial values within the fabrication limited bounds. There are two conditions for the optimization loop to halt: (1) $\vec{p}$ converges, i.e. $\vec{p}_n - \vec{p}_{n-1} < \mathbf{T}$, where $\mathbf{T}$ denotes a threshold, and $n$ represents the iteration number; (2) the number iteration reaches a certain integer $\mathbf{N}$. When one halting conditions is met, the final meta-atom binary structure parameters are converted to GDSII files for fabrication.

## Characterization

We characterized the PSFs of the fabricated meta-optics using a confocal microscopy setup in the LWIR. In this configuration, a tunable, 8.23–10.93 $\mu$m, quantum cascade laser (QCL) with 500 ns pulses and repetition rate of 100 kHz (Daylight Solutions 31090-CT) was incident on a Ge aspheric lens with focal length $f_1 = 20$ mm and NA of 0.63 (Edmund Optics #68-253). This lens focused the QCL onto a 30 $\mu$m diameter pinhole (Thorlabs P30S) where approximately 30% of the incident power was transmitted. The QCL was then incident on the meta-optic, placed $2f_{meta} + f_2 = 35$ mm from the pinhole. The beam was then collimated by a Ge aspheric lens with $f_2 = 15$ mm and NA of 0.83 (Edmund Optics #68-252) placed $f_{meta} + f_2 = 35$ mm after the meta-optic. The beam then traveled to a ZnSe plano-convex lens with $f_3 = 50.8$ mm which focused the beam onto the image plane. This ZnSe lens was created by Rocky Mountain Instrument Co. and had a custom coating covering the 3–12 $\mu$m range with a reflectivity $R_{avg} \leq 5\%$. The image was captured on a strained layer superlattice (SLS) focal plane array (FPA) with a 15 $\mu$m pixel pitch and 640 × 512 pixels (FLIR A6751 SLS) cooled to 76 K. The camera acquisition time was 200 $\mu$s to best fit the 14-bit dynamic range of the FPA and 100 frames were captured and averaged with background correction. The schematic of the setup and the component specifications are present in the Supplementary Materials. The measured PSFs are reported in the Supplementary Materials.

## In-lab imaging

To assess the improved imaging performance of the MTF-engineered meta-optics over the forward-designed metalens, we captured imaged under broadband illumination using black-body radiation from a hotplate with high-emissivity fiberglass tape heated to 150 °C as the light source. Custom aluminum targets were made using laser cutting and finished with matte black paint to prevent reflections. These targets were placed in front of the hotplate, allowing patterned LWIR light to go through, creating contrast. A FLIR Boson 640 camera was placed on the imaging plane of the meta-optic in testing and sent the data to a PC for further post-processing, which included background subtracting, contrast stretching, and block-matching denoising. Through this predefined post-processing routine, we were able to improve the dynamic range and reduce microbolometer array artifacts.

## Solving the deconvolution inverse problem

Figure 4 shows images captured in the wild with our meta-optic. In each case, we deconvolved the measurements by solving an inverse problem using a data-free image prior. Each 640 × 512 image was modeled as output of an implicit neural representation (INR)[31,32] $\mathcal{N}_\theta$ with parameters $\theta$. INRs are continuous functions that map local coordinates ($x, y$ for images) to a given output (image intensity). A unique property enabled by certain INRs, particularly ones equipped with a complex Gabor filter activation function, is the bias for images. This implies that the output tend to look more like images than noise. We leveraged this property to regularize the inverse problem. The

specific inverse problem we solved is

$$\min_{\theta,K} \sum_{n=1}^{N} \sum_{m=1}^{M} \| I_{\text{obs}}(m,n) - (K * \mathcal{N}_\theta)(m,n)\|^2 + \eta_{\text{TV}} \text{TV}(\mathcal{N}_\theta(m,n)), \quad (3)$$

where $I_{\text{obs}}$ is the observed 2D image, $K$ is the PSF of the optical system, TV() is the total variation loss function, and $\eta_{\text{TV}}$ is the weight for the total variation loss. We perform a semi-blind deconvolution where we initialize $K$ to be the analytical PSF from our simulations, and then solve for the parameters of the network and the PSF together.

LWIR sensors based on microbolometers suffer from fixed pattern noise, which causes horizontal and vertical striations. Inspired by recent work on removing fixed pattern noise in thermal images[33], we modeled it as a low-rank image. We then simultaneously solved for the fixed pattern noise, the PSF, and the parameters of the network, giving us

$$\min_{\theta,K,F} \sum_{n=1}^{N} \sum_{m=1}^{M} \| I_{\text{obs}}(m,n) - F(m,n)(K * \mathcal{N}_\theta)(m,n)\|^2 + \eta_{\text{TV}} \text{TV}(\mathcal{N}_\theta(m,n)),$$
$$\text{s.t} \quad \text{rank}(F) = r,$$
$$(4)$$

where $F$ is the fixed pattern noise. We used the recently developed wavelet implicit neural representations (WIRE)[32] for the INR architecture as it resulted in highest qualitative accuracy. Since there is a paucity of high quality thermal images, we found that such deep image prior-based iterative algorithms enable us to obtain high quality reconstructions. As future work, we will evaluate the use of existing pre-trained neural networks and fine-tune them on a small number of thermal images to obtain a feed forward network that will enable real-time reconstruction. An overview of the reconstruction pipeline is shown in Supplementary Fig. 5.

## Fabrication

The meta-optics are fabricated on a 500 $\mu$m thick double-side polished silicon wafer, lightly doped with boron, giving a sheet resistivity of 1–10 $\Omega$·cm. Direct-write lithography (Heidelberg DWL 66+) defines the aperture locations in the photoresist covering the wafer surface. A 220 nm thick aluminum layer is deposited onto the patterned photoresist via electron beam evaporation (CHA Solution) and lifted off to construct the metal mask around the circular apertures, helping to reduce noise during the experiments. The wafer is coated with another photoresist and patterned with the meta-optic scatterers via direct-write lithography, aligned inside the defined apertures. The photoresist pattern of meta-optics is transferred to the bulk silicon to a depth of 10 $\mu$m by deep reactive ion etching (SPTS DRIE), utilizing its capability to etch with high aspect ratios and vertical sidewalls. After etching, the photoresist residue is stripped, and the fabricated meta-optics are ready for characterization. This process is shown on Supplementary Fig. 10.

## Data availability

The datasets analyzed during the current study are available at https://zenodo.org/records/10400870.

## Code availability

The differentiable meta-optics inverse design software package is available at: https://github.com/luochenghuang/metabox[34].

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

## Acknowledgements

Funding for this work was supported by the federal SBIR program. Part of this work was conducted at a National Nanotechnology Coordinated Infrastructure (NNCI) site at the University of Washington with partial support from the National Science Foundation. J.R.H. acknowledges support from the Air Force Office of Scientific Research (Program Manager Dr. Gernot Pomrenke) under award number FA9550-20RYCOR059.

## Author contributions

A.M. and L.H. conceived the idea. L.H. developed the metasurface simulation and optimization pipeline, led the experiment, and authored the manuscript. Z.H. was responsible for fabricating all the devices. A.W.S. and S.M. assisted with in-lab imaging. V.S., S.M., and Z.H. performed the ambient Long-Wavelength Infrared (LWIR) imaging. V.S. was the developer of the deconvolution algorithm. J.F. contributed to the manuscript writing. Q.T. created the 3D illustration of the metasurface. J.R., R.G., J.H., P.H., O.K., and Z.C. were in charge of conducting the Point Spread Function (PSF) measurements. A.M., A.V., and K.B. provided project supervision and editorial assistance for the manuscript.

## Competing interests

A.M. and K.B. are co-founders of Tunoptix, which is trying to commercialize relevant technology. Z.C. works at CFDRC who is commercializing thermal imaging systems. The remaining authors declare no competing interests.
