## [Peer Review File · Nature Communications]

Broadband Thermal Imaging using Meta-OpticsReviewer #1 (Remarks to the Author):

This paper presents broadband LWIR metalenses for thermal imaging. A MTF-based optimization procedure was applied. The design was validated by experimental data. The whole paper is technically sound. But there are a few additional comments that need to be addressed and clarified.

1. One prominent consideration throughout the paper revolves around the accuracy of the PSF and Strehl Ratio calculation method.

1a) A comparison between the designs in Fig. 1 and SEM images in Fig. 2b indicates a potential design-to-fabrication gap. The precision of the calculated transmission and phase delay via RCWA warrants clarification.

1b) Considering that RCWA pertains to periodic structures, the PSF calculated using element-wise performance, as seen in [M. V. Zhelyeznyakov et al., "Deep Learning to Accelerate Scatterer-to-Field Mapping for Inverse Design of Dielectric Metasurfaces," ACS Photonics, 2021], raises questions about its reliability.

1c) the accuracy and experimental validation of the "Shifted angular spectrum propagation" method require discussion. To enhance the credibility of the optimization framework, it would be prudent for the authors to present measured Strehl ratios and compare them with the results depicted in Fig. 1d.

2. Other approaches, such as that presented by [S. Shrestha et al., "Broadband achromatic dielectric metalenses," Light: Science & Applications, vol. 7, no. 1, pp. 1-11, 2018], aim to maximize efficiency across a wide band. Although these approaches are subject to limitations in meta-atom thickness and lens diameter, their overall efficiency remains notably high. Given that the paper's optimization framework concentrates solely on MTF, it would be valuable to address concerns regarding potential efficiency trade-offs. Particularly, could the efficiency suffer, and could this eventually pose issues for detectors, particularly when lens diameters are substantially reduced?

3. In Section 4.5, the authors mentioned that K is the PSF of the optical system from simulation. Can the authors further clarify how K was calculated analytically?

Reviewer #2 (Remarks to the Author):

In this work, Huang et al., demonstrated an all-silicon polychromatic meta-optics for broadband thermal imaging in the LWIR ranging from 8 to 12 micron. The meta-optics is optimized by using a differentiable inverse design framework, and the objective function is the wavelength-averaged volume under the modulation transfer function (MTF). The design goal is uniform Strehl ratios at discrete wavelengths spanning the working bandwidth, which are assumed helpful to mitigate the chromatic aberration and improve the broadband imaging quality. The imaging demo shows the comparison between hyperboloid and MTF-engineered metalens comprising simple scatters and complex scatters. The meta-optics using complex scatters and MTF-engineering design strategies show better performance than the other two, especially under different bandpass filters. Furthermore, the authors demonstrated "in the wild" imaging, where pronounced differences between MTF-engineered meta optics and Hyperboloid meta optics are observed. Considering the novelty of the design methodology and demonstration of meta-optics in broadband thermal imaging, I recommend the work to be considered for publication. Here below are some comments for further improvement of the manuscript:

1. Could the authors explain why the Strehl ratio of the forward-designed hyperboloid metalens has a very low Strehl ratio of only 0.12 in Fig. 1b? And what is the angle of incidence when defining the Strehl ratio? Judging from data presented in Fig. S2, I expect a much higher Strehl ratio of the hyperboloid metalens (at the design wavelength) compared to the MTF-engineered ones.

2. From a physics point of view, could the authors explain why complex-shaped meta atoms have better performance than the simple ones? Is it partially contributed by the dispersion engineering capabilities by the complex meta atoms?

3. Could the authors comment on whether efficiency is an issue in the case of broadband thermal imaging? Since polychromatic design will sacrifice efficiency a lot, especially when increasing the number of design wavelengths.

Some minor comments:

4. What is Manhattan grid? Do you mean by a square lattice?

5. Throughout the design flow, angle of incidence is shown as a variable. Could the authors give information on how many angles are considered in calculating the PSFs?

6. Could the authors comment on computational cost of the image deconvolution method? Does it support video framerate?

7. Is the shutter correction in the image processing flow a physical (or experimental) correction?

8. Typos: title for Fig.3, "Hyperboloid" instead of "Hyperbloid". Figure captions for Fig. S2.

Response letter to the reviewers of the article “**Broadband Thermal Imaging using Meta-Optics**”

First, we would like to thank the reviewers for their time and effort to carefully read our paper and provide us with constructive feedback. Their comments have certainly helped us improve the manuscript. We appreciate the reviewers for considering our work as “*technically sound*” and “*novel*”. We have also taken the criticisms seriously and modified our manuscript to take account of all the suggestions from the reviewers. Below you can find our point-by-point replies to all the reviewers’ remarks. The changes are highlighted in red in the revised manuscript.

Reviewer #1 (Remarks to the Author):

This paper presents broadband LWIR metalenses for thermal imaging. A MTF-based optimization procedure was applied. The design was validated by experimental data. The whole paper is technically sound. But there are a few additional comments that need to be addressed and clarified.

Response: We thank the reviewer for carefully reading our paper and their positive assessment of our paper. Below are our responses to the comments/ criticisms raised by the reviewer.

1. One prominent consideration throughout the paper revolves around the accuracy of the PSF and Strehl Ratio calculation method.

1a) A comparison between the designs in Fig. 1 and SEM images in Fig. 2b indicates a potential design-to-fabrication gap. The precision of the calculated transmission and phase delay via RCWA warrants clarification.

Response: We appreciate the reviewer's insightful comments regarding the calculation of the PSF and Strehl ratio. We acknowledge that there indeed is a design-to-fabrication discrepancy that the reviewer identified between Fig. 1 and the SEM images in Fig. 2b.

To address this, we have conducted further analyses to ascertain the precision of the calculated transmission and phase delay via rigorous coupled wave analysis. We have incorporated additional data points in the paper to demonstrate the robustness of our calculations.

1. *We have computed the phase and transmission difference for pillars with varying dimensions (to assess the effect of inevitable fabrication imperfections). This helps in understanding the deviation from the ideal design when the size of the pillars differs from the intended design. This analysis will shed light on the sensitivity of our method to fabrication inaccuracies.*
2. *To visualize the deviation from the ideal design, we have utilized scatter plots. The designed value is represented along the x-axis and the shifted value along the y-axis. Ideally, all data points should align on a straight line, indicating that the fabricated design matches the calculated design. The correlation coefficient (R-squared value) is calculated to quantify the accuracy of the design to fabrication process.*

*These additional analyses and results are now included in the revised supplemental materials to provide more comprehensive insights into the reliability of our meta-atom transmission calculations and better address the concerns raised by the reviewer. However, we would like to point out that our computational backend can help circumvent these unavoidable fabrication errors and the subsequent deviation in the desired phase distribution. We have now added this aspect also in the main text: **“We note that, in these simulations, we added normally distributed perturbation into each meta-atom’s design parameters, simulating fabrication imperfections, resulting in less-than-perfect Strehl ratio for the hyperboloidal metasurface at the desired wavelength. More details on the effect of fabrication imperfections on the properties of meta-atoms are reported in the supplementary materials. We emphasize, however, while fabrication imperfections will affect the meta-optics captured images, the use of computational backend can provide additional robustness in the overall imaging performance.”***

1b) Considering that RCWA pertains to periodic structures, the PSF calculated using element-wise performance, as seen in [M. V. Zhelyeznyakov et al., "Deep Learning to Accelerate Scatterer-to-Field Mapping for Inverse Design of Dielectric Metasurfaces," ACS Photonics, 2021], raises questions about its reliability.

Response: We thank the reviewer for their insightful comments concerning the RCWA simulations and the potential limitations associated with the element-wise performance calculation/ locally periodic approximation. We agree with the point about the importance of considering near-field coupling between adjacent scatterers to enhance simulation accuracy.

However, we would like to clarify that the method employed by M. V. Zhelyeznyakov et al., which uses a physics-informed neural network (PINN), is currently only applicable predominantly to 2D scatterers. Our study, on the other hand, deals with 3D structures. Therefore, directly applying their method to our case would be challenging.

Additionally, the complexity of our scatterers, which possess 3 degrees of freedom each, significantly increases the computational time required by such a method. This time intensifies when considering the additional 3 degrees of freedom for the neighboring scatterers.

Nonetheless, we have found that the Local Periodic Approximation (LPA) provides a reasonable accuracy for our purposes, as evident in our earlier broadband imaging results in the visible. We do acknowledge that going beyond LPA, can improve the efficiency, however, that is a minor improvement over what can be achieved under LPA. We understand the concern about the credibility of our simulation methodology and appreciate your suggestion to validate our method. In fact, apart from a full-wave simulation, any other approach of simulating meta-optics will invariably use some sort of approximation. We found LPA to be a very good approximation for imaging applications. We also emphasize that the use of computational backend likely has allowed the detrimental effect of such approximation to be minimal.

1c) the accuracy and experimental validation of the "Shifted angular spectrum propagation" method require discussion. To enhance the credibility of the optimization framework, it would be prudent for the authors to present measured Strehl ratios and compare them with the results depicted in Fig. 1d.

Response: We appreciate the reviewer's suggestion to discuss the accuracy and experimental validation of the "Shifted Angular Spectrum Propagation" method to fortify the credibility of our optimization framework.

In response to this, we have referenced several studies that have employed the same method and have presented experimental results. These papers provide robust evidence supporting the accuracy and practicality of the method. They include:

- *Neural nano-optics for high-quality thin lens imaging*
 - <https://www.nature.com/articles/s41467-021-26443-0>

- *Photonic Advantage of Optical Encoders (point to the supplementary)*

- <https://arxiv.org/ftp/arxiv/papers/2305/2305.01743.pdf>

Additionally, to further validate our approach, we have added the experimental measurements of the Strehl ratios and compared them with the results demonstrated in Fig. 1d. This comparison further corroborates the accuracy of our design and optimization framework, and these results have been added to the revised manuscript.

We hope that these additions address the reviewer's queries and provide additional confidence in the reliability of our methodology.

2. Other approaches, such as that presented by [S. Shrestha et al., "Broadband achromatic dielectric metalenses," *Light: Science & Applications*, vol. 7, no. 1, pp. 1-11, 2018], aim to maximize efficiency across a wide band. Although these approaches are subject to limitations in meta-atom thickness and lens diameter, their overall efficiency remains notably high. Given that the paper's optimization framework concentrates solely on MTF, it would be valuable to address concerns regarding potential efficiency trade-offs. Particularly, could the efficiency suffer, and could this eventually pose issues for detectors, particularly when lens diameters are substantially reduced?

Response: We thank the reviewer for their insightful comments. In metalens community, historically two different efficiencies have been employed: Transmission and Focusing efficiency. The transmission efficiency tells us how much light gets transmitted through the optic, and focusing efficiency determines how much of those transmitted light gets in the focused region. However, we found these efficiencies to be somewhat "made-up" and no prior precedence exists on this beyond meta-optics community. No refractive optics defines their focusing efficiency. Moreover, the extent of the focal spot is arbitrary. There is no justification why three times the full-width half maxima will be used to measure the focusing efficiency. In fact, several papers, including the paper by Shrestha et al., defines this by "three to five" FWHM, and based on our own experience, experimentally controlling a pinhole to achieve "three to five" FWHM for a small metalens is highly noisy!! We believe this introduction of "focusing efficiency" stems from the use of diffraction efficiency in multi-level diffractive optics, where often we want most of light to go to a specific diffraction order. Given these diffraction orders are far more separated, such a definition makes sense, and can be measured with high accuracy. Given in metalens (at least what we have

presented here), there is no higher order diffraction, all our light (~100% of the transmitted light) ends up in zero-th order diffraction. Of course, this does not mean we have 100% “focusing efficiency”!! Given we are using computational backend, such focusing efficiency is also not of high relevance, as long as we can measure the information from the scene on the camera with high signal-to-noise ratio. Given, the community is seriously asking whether a meta-lens can outperform refractive, we believe we should move away from measuring a noisy “focusing efficiency”!! Transmission efficiency is, however, very important, and much easier to measure with high accuracy. For most meta-optics, including the one we have presented in this paper, the transmission efficiency is greater than 60%, which is sufficient for many applications.

This brings us to the second part of reviewer’s point: does our MTF engineering approach consider efficiency? Our MTF engineering aims to maximize the volume under the MTF curve. We define our “modified Strehl ratio” as the ratio of volume under the calculated MTF curve (without any normalization) and a diffraction limited MTF curve. This ratio drops if the light is not well focused. A large amount of scattered light will add a large DC component to the MTF, and the overall volume under the curve goes down. Thus, our MTF engineering approach indirectly considers the “focusing efficiency”. We emphasize that, using MTF also gives us a way to directly compare with the commercial refractive optics. We note that, the seemingly high Strehl ratio/focusing efficiency from other dispersion engineering paper is simply because they have extremely small aperture compared to ours. With that small aperture, our method also gets ~80% Strehl ratio.

Our method, however, does neglect transmission efficiency. This efficiency is of course very important, and also gives a direct comparison point with the refractive. We found that this efficiency can be estimated from the meta-atoms themselves, and as such currently implemented as a pre-selection step to ensure only high transmission (>60%) meta-atoms are used for design. One way to implement them in the MTF engineering framework is by modelling the complex phase transmission coefficient, rather than pure phase modulation. We have added this discussion in the main text as follows: **“To ensure high transmission efficiency, we retained only those meta-atoms which have transmission exceeding 60%.”**

We also have added this in the paper: **“We note that, unlike many other works, we have not directly emphasized the need for high efficiency. In meta-optics community, historically two**

different efficiencies have been reported: transmission and Focusing efficiency. The transmission efficiency indicates how much light gets transmitted through the optic, and focusing efficiency determines how much of the transmitted light gets into the focused region. The focusing efficiency is somewhat arbitrarily defined and has almost no counterpart for refractive optics. Hence, in our work, we do not optimize focusing efficiency. However, our modified Strehl ratio implicitly takes account of the focusing efficiency. If the light is not tightly confined, and a large amount of scattered light is present, we will have a large DC component in the MTF which will reduce the average Strehl ratio. Thus, our MTF engineering method indirectly optimizes the focusing efficiency. To ensure high transmission efficiency, we pre-select the meta-atoms with high transmission coefficient.”

3. In Section 4.5, the authors mentioned that K is the PSF of the optical system from simulation. Can the authors further clarify how K was calculated analytically?

Response: We assumed the lens is designed for 10 μ m, and the PSF is Gaussian. Then we calculated the PSF for other wavelengths, assuming a hyperbolic phase function. Although this method only provides a rough estimate of the PSF, we have empirically found it to be a good candidate. As such, we found that even theoretical PSF provides high-quality reconstruction. This is important especially in LWIR range, where the experimentally measured PSFs are often very noisy.

Reviewer #2 (Remarks to the Author):

In this work, Huang et al., demonstrated an all-silicon polychromatic meta-optics for broadband thermal imaging in the LWIR ranging from 8 to 12 micron. The meta-optics is optimized by using a differentiable inverse design framework, and the objective function is the wavelength-averaged volume under the modulation transfer function (MTF). The design goal is uniform Strehl ratios at discrete wavelengths spanning the working bandwidth, which are assumed helpful to mitigate the chromatic aberration and improve the broadband imaging quality. The imaging demo shows the comparison between hyperboloid and MTF-engineered metalens comprising simple scatters and complex scatters. The meta-optics using complex scatters and MTF-engineering design strategies show better performance than the other two, especially under different bandpass filters. Furthermore, the authors demonstrated “in the wild” imaging, where pronounced differences

between MTF-engineered meta optics and Hyperboloid meta optics are observed. Considering the novelty of the design methodology and demonstration of meta-optics in broadband thermal imaging, I recommend the work to be considered for publication. Here below are some comments for further improvement of the manuscript:

Response: We thank the reviewer for carefully reading our paper and their positive assessment of our paper. Below are our responses to the comments/ criticisms raised by the reviewer.

1. Could the authors explain why the Strehl ratio of the forward-designed hyperboloid metalens has a very low Strehl ratio of only 0.12 in Fig. 1b? And what is the angle of incidence when defining the Strehl ratio? Judging from data presented in Fig. S2, I expect a much higher Strehl ratio of the hyperboloid metalens (at the design wavelength) compared to the MTF-engineered ones.

*Response: The reviewer is correct that a hyperboloid lens at a single wavelength and normal incidence is expected to give very high Strehl ratio. There are several reasons, why our Strehl ratio is 0.12. First of all, our Strehl ratio is slightly different from the traditional definition. As we are defining the Strehl ratio as the ratio between the volume under the MTF curve, our Strehl ratio implicitly considers the focusing efficiency. Hence, we have slightly lower Strehl ratio than what one expects from a hyperboloid. Additionally, we introduced normally distributed perturbation into each meta-atom's design parameters, to mimic fabrication imperfections, resulting in less-than-perfect Strehl ratio for the hyperboloidal metasurface. We have added this tandem to the figure description for better readability. Furthermore, we have added the ideal Strehl ratios in the supplement (supplementary material section 9). We have also added the following section to the main text for better clarity: **"We note that, in these simulations, we added normally distributed perturbation into each meta-atom's design parameters, simulating fabrication imperfections, resulting in less-than-perfect Strehl ratio for the hyperboloidal metasurface at the desired wavelength. More details on the effect of fabrication imperfections on the properties of meta-atoms are reported in the supplementary materials. We emphasize, however, while fabrication imperfections will affect the meta-optics captured images, the use of computational backend can provide additional robustness in the overall imaging performance."***

2. From a physics point of view, could the authors explain why complex-shaped meta atoms have better performance than the simple ones? Is it partially contributed by the dispersion engineering capabilities by the complex meta atoms?

*Response: The reviewer raised an important point. We do believe that having more degrees of freedom on parameterization of meta-atoms opens doors to more possible designs and more supported modes. As such, the broadened design space would better accommodate the merit function. As the reviewer pointed out, this is analogous to having more meta-atom candidates to cover more phase delay, group delay, and group delay dispersion diversity in dispersion engineered meta-optics. We have now added this analogy to the discussion of the main text: **“We can qualitatively explain the higher Strehl ratio with complex scatterers, as they can provide higher phase diversity, which will help to satisfy the phase distribution for different wavelengths. Essentially, such complex scatterers help to achieve a similar effect of dispersion engineering to achieve broadband performance.”***

3. Could the authors comment on whether efficiency is an issue in the case of broadband thermal imaging? Since polychromatic design will sacrifice efficiency a lot, especially when increasing the number of design wavelengths.

Response: We thank the reviewer for raising this concern on efficiency. We note that the other reviewer raised a similar concern as well. In metalens community, historically two different efficiencies have been employed: Transmission and Focusing efficiency. The transmission efficiency tells us how much light gets transmitted through the optic, and focusing efficiency determines how much of those transmitted light gets in the focused region. However, we found these efficiencies to be somewhat “made-up” and no prior precedence exists on this beyond meta-optics community. No refractive optics defines their focusing efficiency. Moreover, the extent of the focal spot is arbitrary. There is no justification why three times the full-width half maxima will be used to measure the focusing efficiency. In fact, several papers, including the paper by Shrestha et al., defines this by “three to five” FWHM, and based on our own experience, experimentally controlling a pinhole to achieve “three to five” FWHM for a small metalens is highly noisy!! We believe this introduction of “focusing efficiency” stems from the use of diffraction efficiency in multi-level diffractive optics, where often we want most of light to go to a specific diffraction order. Given these diffraction orders are far more separated, such a definition makes sense, and can be

measured with high accuracy. Given in metalens (at least what we have presented here), there is no higher order diffraction, all our light (~100% of the transmitted light) ends up in zero-th order diffraction. Of course, this does not mean we have 100% “focusing efficiency”!! Given we are using computational backend, such focusing efficiency is also not of high relevance, as long as we can measure the information from the scene on the camera with high signal-to-noise ratio. Given, the community is seriously asking whether a meta-lens can outperform refractive, we believe we should move away from measuring a noisy “focusing efficiency”!! Transmission efficiency is, however, very important, and much easier to measure with high accuracy. For most meta-optics, including the one we have presented in this paper, the transmission efficiency is greater than 60%, which is sufficient for many applications.

In our case, the transmission efficiency does not strongly depend on the polychromatic approach. i.e., how many wavelengths we are using. Our transmission efficiency is largely governed by the choice of meta-atoms. However, the modified Strehl ratio does depend on the number of wavelengths being used. With a greater number of wavelengths being optimized, such average Strehl ratio will change. We do not have a definitive answer in terms of final limit of the Strehl ratio, but anecdotally, as we increase the number of design wavelengths, we expect to see the Strehl ratio at each wavelength will start to become similar but will be lower than the highest Strehl ratio that is achievable in a particular wavelength if the lens is optimized only at that wavelength. However, we emphasize that, as long as we preserve high spatial frequency components, we can reconstruct large part of the image using a computational backend.

As such, we have added a section to the discussion section of the main text, which reads as follows: “We note that, unlike many other works, we have not directly emphasized the need for high efficiency. In meta-optics community, historically two different efficiencies have been reported: transmission and Focusing efficiency. The transmission efficiency indicates how much light gets transmitted through the optic, and focusing efficiency determines how much of the transmitted light gets into the focused region. The focusing efficiency is somewhat arbitrarily defined and has almost no counterpart for refractive optics. Hence, in our work, we do not optimize focusing efficiency. However, our modified Strehl ratio implicitly takes account of the focusing efficiency. If the light is not tightly confined, and a large amount of scattered light is present, we will have a large DC component in the MTF which will reduce the average Strehl ratio. Thus, our MTF-

engineering method indirectly optimizes the focusing efficiency. To ensure high transmission efficiency, we pre-select the meta-atoms with high transmission coefficient.”

Some minor comments:

4. What is Manhattan grid? Do you mean by a square lattice?

Response: We appreciate the reviewer for the suggestion. We now have adopted “square lattice” in our manuscript:

*“The meta-atoms sit on a **square lattice** with a periodicity Λ set to $4 \mu\text{m}$.”*

5. Throughout the design flow, angle of incidence is shown as a variable. Could the authors give information on how many angles are considered in calculating the PSFs?

Response: We used 0 and 10 degrees for optimization, which is now mentioned in section 4.2 of the main text. We apologize for missing it.

6. Could the authors comment on computational cost of the image deconvolution method? Does it support video framerate?

*Response: In our work we have employed two different computational methods. For indoor imaging (main text) and outdoor imaging (the ones reported in the supplement), we employed Weiner Deconvolution and bm3D algorithm. They are computationally simple and can provide video capture. We have verified this in the visible wavelength range (publication under preparation). We also used a more complex iterative algorithm, which does provide higher signal-to-noise ratio performance for the outdoor imaging (reported in the main text). This method being iterative, cannot readily support video rate imaging. However, we can train a feed forward neural network (future work) for real time performance. We have clarified this in the modified manuscript and added this: **“Since there is a paucity of high quality thermal images, we found that such deep image prior-based iterative algorithms enable us to obtain high quality reconstructions. As future work, we will evaluate the use of existing pre-trained neural networks and fine-tune them on a small number of thermal images to obtain a feed forward network, that will enable real-time reconstruction.”***

7. Is the shutter correction in the image processing flow a physical (or experimental) correction?

Response: It is a physical process. Immediately after each image capture, the shutter is closed momentarily, allowing us to record the background noise floor of the sensor. This correction is important as the microbolometer arrays have time varying noise floor.

8. Typos: title for Fig.3, “Hyperboloid” instead of “Hyperblloid”. Figure captions for Fig. S2.

Response: We thank the reviewer for the feedback. We have corrected the typo. We also carefully read the whole paper to correct any other typos and mistakes.

Reviewer #1 (Remarks to the Author):

The authors have addressed my comments.

Reviewer #2 (Remarks to the Author):

Dear Authors

Thanks for addressing my comments point to point. The revised manuscript contains more technical details and comments. I recommend the manuscript to be considered for publication.

One optional comment for your future consideration (you do not need to address this for revision of current manuscript). Using MTF volume as figure of merit is kind of tricky when optimizing the high performance imaging component. In our previous experiments, we found the case that a optimized single wavelength lens show ghost foci, due to fabrication imperfection and thus unwanted scattering. However, the MTF of the design focal spot is close to diffraction limited curve, and the strehl ratio is above 0.9. Its focusing efficiency is less than 50%, and the metalens shows strong background noise when performing even single-wavelength imaging. In my opinion, high transmission of meta atoms is necessary but not sufficient for high-performance meta-optics, especially when comparing it with refractive counterpart. If the strehl ratio defined in the work contains the information of "focusing efficiency", then I expect its value will decrease as increasing the number of wavelengths to be optimized. That is being said, there is trade-off between wavelength channels vs. signal to noise ratio per channel. At the same time, I agrees with the authors that as long as the device shows reason performance in real-world application, the proposed design strategy is valuable.

One optional comment for your future consideration (you do not need to address this for revision of current manuscript). Using MTF volume as figure of merit is kind of tricky when optimizing the high performance imaging component. In our previous experiments, we found the case that an optimized single wavelength lens shows ghost foci, due to fabrication imperfection and thus unwanted scattering. However, the MTF of the design focal spot is close to diffraction limited curve, and the strehl ratio is above 0.9. Its focusing efficiency is less than 50%, and the metalens shows strong background noise when performing even single-wavelength imaging. In my opinion, high transmission of meta atoms is necessary but not sufficient for high-performance meta-optics, especially when comparing it with refractive counterpart. If the strehl ratio defined in the work contains the information of "focusing efficiency", then I expect its value will decrease as increasing the number of wavelengths to be optimized. That is being said, there is trade-off between wavelength channels vs. signal to noise ratio per channel. At the same time, I agree with the authors that as long as the device shows reason performance in real-world application, the proposed design strategy is valuable.

Response: We thank the reviewer for the comment and suggestions. In our calculation of the modulation transfer function, we take a direct Fourier Transformation of the spatial intensity in the focal plane. In such analysis, the background light will create a large DC component and thus MTF will suffer. However, if the MTF is calculated by subtracting the background (often in PSF measurements), then indeed a situation can arise with diffraction limited MTF, and high Strehl ratio (based on the amplitude of the PSF), but low focusing efficiency. The reviewer is also correct that there is a trade-off between the number of wavelength channel and SNR per channel. In our case we optimized over only ~8 channels due to limited computational resources. With enhanced computational resources, one can explore the fundamental limit in terms of performance, which is what we are currently working on.